# Study on the effect of digital economy on high-quality economic development in China

**Wei Zhang, Siqi Zhao**  **\*, Xiaoyu Wan, Yuan Yao**

School of Economics and Management, Chongqing University of Posts and Telecommunications, Chongqing, China

\* 1104271840@qq.com

**Data Availability Statement:** The data underlying the results presented in the study are available from http://www.stats.gov.cn/. The data results in the manuscript and the data sets in the supporting information are all collected by authors.

## Abstract

At present, the digital economy, which takes information technology and data as the key elements, is booming and has become an important force in promoting the economic growth of various countries. In order to explore the current dynamic trend of China's digital economy development and the impact of the digital economy on the high-quality economic development, this paper measures the digital economic development index of 30 cities in China from the three dimensions of digital infrastructure, digital industry, and digital integration, uses panel data of 30 cities in China from 2015 to 2019 to construct an econometric model for empirical analysis, and verifies the mediating effect of technological progress between the digital economy and high-quality economic development. The results show that (1) The development level of China's digital economy is increasing year by year, that the growth of digital infrastructure is obvious, and that the development of the digital industry is relatively slow. (2) Digital infrastructure, digital industry and digital integration all have significant positive effects on regional total factor productivity, and the influence coefficients are 0.2452, 0.0773 and 0.3458 respectively. (3) Regarding the transmission mechanism from the digital economy to the high-quality economic development, the study finds that the mediating effect of technological progress is 0.1527, of which the mediating effect of technological progress in the eastern, northeast, central and western regions is 1.70%, 9.25%, 28.89% and 21.22% respectively. (4) From the perspective of spatial distribution, the development level of the digital economy in the eastern region is much higher than that in other non-eastern regions, and the development of digital economy in the eastern region has a higher marginal contribution rate to the improvement of the total factor productivity. This study can provide a theoretical basis and practical support for the government to formulate policies for the development of the digital economy.

## 1 Introduction

China's fourteenth five-year plan for national economic and social development proposes to build a new development pattern of domestic and international dual cycles. Accelerating the construction of a new development pattern with the domestic big cycle as the main body and the domestic and international double cycles mutually promoting each other is an important

**Funding:** This work was supported by the Project of Humanities and Social Sciences Ministry of Education in China [18YJC790224], the Ministry of Education Layout Foundation of Humanities and Social Sciences [19XJA630004], the Planning Project of Chongqing Social Science [2016BS057], the talent Introduction Project of Chongqing University of Posts and Telecommunications [A2016-04, K2015-128]. The funders play a guiding and supporting role in our study, and they had no role in study design, data collection and analysis, decision to publish, or preparation of the manuscript.

**Competing interests:** No authors have competing interests

strategic choice for connecting domestic and international markets, promoting high-quality development of an open economy, and adapting to new changes in the global economy. The digital economy, dominated by a new generation of information technology, has become an important path for efficiently meeting market demand and smoothing the domestic and international dual cycles, and will become an important driving force and breakthrough point for boosting the new development pattern of dual cycles. In recent years, the digital economy has grown rapidly in China. As a new economic form that leads the future, it has unprecedentedly reconstructed a new picture of economic and social development and is a new variable for improving economic quality and efficiency. According to the data of the 2019 Digital Economy Report released by the United Nations International Trade, based on the report's definition of the digital economy, the scale of the digital economy accounts for between 4.5% and 15.5% of the world's gross domestic product (GDP). In terms of the added value of information and communication technology, the United States and China account for 40% of the world's total added value, and the development of the digital economy has become a common choice for major global powers and regions to reshape global competitiveness. According to the most recent "White Paper on the Development of China's Digital Economy (2021)" released by the China Academy of Information and Communications Technology, the proportion of China's digital economy in GDP has increased year by year from 2005 to 2020, from 14.2% to 38.6%.

At present, a unified standard has not yet been setup for the academic definition of the concept of the digital economy. In the academic field, in 1996, Tapscott, an American IT consulting expert, first put forward the concept of a digital economy in the report "Digital Economy: Opportunities and Risks in the Era of Network Intelligence". The concept's main feature is the digital flow and transmission of information over the network [1]. In 1998, in the government report "Emerging Digital Economy" released by the US Department of Commerce, the term digital economy was included for the first time, and the concept of the digital economy was gradually recognized by governments and scholars worldwide. Since then, the related research on the digital economy has begun to rise, the concept of the digital economy has been continuously enriched and deepened in this process, and the research category of the digital economy has been constantly improved. There are two main points of view to define the digital economy in a narrow sense. One is that the digital economy is divided into two parts, namely, ICT services and manufacturing, which are summed up as the digital economy, and the other part comprises retail, the platform economy and the sharing economy, which are mainly supported by ICT and cannot be distinguished by official industry codes. Maglio believes that the digital economy consists of four parts: Internet infrastructure, e-commerce, the digital delivery of goods and services, and the retail sales of tangible goods. Meisenberg believes that the digital economy has three main components: the e-business infrastructure, e-business and e-commerce. In recent years, many studies have further defined related products or industries on the basis of defining the components of the digital economy. Bukht divides the digital economy into three levels: the first layer is the core layer, which is the digital (IT/ICT) domain, including hardware manufacturing, software and IT consulting, information services, and telecommunications. The second layer is the narrow caliber, which is the narrow digital economy, including e-commerce, digital services, and platform economy. The third layer is the wide caliber, that is, the broad sense of digital economy, including e-commerce, Industry 4.0, precision agriculture, and the algorithm economy [2]. With the development digital technology, including emerging digital technology and the resulting new industries and new business types, the connotation of the digital economy has been gradually enriched. As a historical category, in the process of the interaction of technology, organization and institution in the economic system, the digital economy is a macro emergence, which is based on the high coordination of human economic

activities and the continuous optimization of the new production organization mode shaped by the high coordination of human economic activities and an interaction based on technology [3, 4]. With the in-depth advancement of supply-side structural reforms, digital technology has flourished and rapidly innovated, and has penetrated widely into other economic fields. Chinese scholars generally divide the digital economy into digital industrialization and industrial digitization. Digital industrialization is equivalent to the traditional information industry, including electronic and communication equipment manufacturing, Internet-related services, software and information technology services in the national economic industry classification. Due to the continuous integration of information technology and other industrial sectors of the national economy, the added value of the digital economy is generated in traditional industries. This portion represents the part of the digitalization of the industry.

At present, the research on the domestic digital economy focuses on the discussion of the development strategy of China's digital economy and its comparison with other countries (regions). Based on the basic connotation of the digital economy, with the deepening of digitalization, scholars believe that the combination of digital technology and traditional industries can achieve the following: realize the green and rapid development of the GDP [5, 6]; realize the transformation of the consumption structure [7]; improve the quality of human capital [8, 9]; drive the industrial economy from being labor-intensive to becoming technology-intensive; consolidate infrastructure construction [10–12]; make full and effective use of data resources [13, 14]; strengthen technological innovation; deepen integrated applications; and create a relaxed environment to better develop the digital economy. China should strengthen its advantages in areas such as 5G, in global digital competition, further strengthen the research and development of key core technologies and develop the whole industry chain. Other scholars believe that in the development of China's digital economy there are some problems, such as unbalanced, inadequate and uncoordinated development. These problems are mainly concentrated in the areas comprising the digital infrastructure, the degree of upgrading of the digital industry, information network security, and the deep integration of digital technology penetration and traditional industries. There is a phenomenon of digital poverty accumulation in space [15]. At present, China's economy is in a period of transitional development and slow growth, and the development of the economy depends more on the quality and efficiency of economic growth than on quantity and speed [16]. The quantitative indicators of economic growth, such as gross domestic product and national income, are no longer the only focus of the government, as the focus has gradually shifted from quantity to quality in order to promote the construction of resource-conserving and environmentally friendly societies [17]. Starting with the economic growth theory of classical economics, the primary concern of economists has always been the quantity rather than the quality of economy's growth. With the help of the Moran index and a spatial econometric model, Bai et al. pointed out that there are significant spatial spillover benefits in China's economic growth and that the economic growth of a province depends on both local input and neighbouring regional input [18]. To provide a basis for a clear understanding of the source of economy growth, other scholars use various econometric models to explore the specific effects of driving factors such as education, political uncertainty, foreign direct investment, human capital, banking globalization and information and communication technology, on the quantitative indicators of economic growth [19–21]. Developing an evaluation index system of high-quality economic development, Qi constructs an economy growth quality measurement system covering scale, performance, structure and coordination [22]. Based on the matrix method, Frolov constructs a regional economic growth quality evaluation system that combines the average annual productivity growth rate and per capita development index [23]. Regarding the research on the relationship between the digital economy and high-quality economic development, more scholars have studied the impact of

the digital economy on high-quality economic development from the perspectives of big data empowerment [24], the integration of the digital economy and the real economy [25], shared economy [26], digital finance [27], and policy supply systems [28]. Generally, the digital economy is in line with the new development concept of innovation, coordination, green, openness and sharing, and in China is becoming a major strategic development direction, which will effectively promote the high-quality development of China's economy.

From the above research results, it can be found that the research contents of scholars in the field of digital economy are relatively extensive, involving the connotation, characteristics, impact effects, index system construction and evaluation and other aspects. However, at present, there are few empirical studies on the promotion of high-quality economic development by digital economy, and few literatures use panel data to study the dynamic changes of the development of digital economy in China, which leads to the limitations of the evaluation of the index system and the lack of continuity in the observation of the development of digital economy. On the one hand, on the basis of previous studies on the index system of the digital economy, this paper selects the core dimensions and measurement indicators of the digital economy, and tries to take the development of the interprovincial digital economy in China from 2015 to 2019 as the research object in order to evaluate the overall changes and regional differences in the development of China's digital economy. On the other hand, on the basis of the evaluation index of the digital economy, an econometric model is constructed to study the effect of the digital economy on high-quality economic development, and technological progress is introduced as an intermediary variable between the two aspects, developing thereby a more objective study of the influence mechanism of the digital economy on high-quality economic development and an analysis from a new perspective. Additionally, this paper also examines the interaction between the digital economy and interprovincial factors to study the regional heterogeneity of the digital economy in promoting high-quality economic development. Moreover, this study deeply explains that the impact of the digital economy on high-quality economic development in the eastern region has a higher marginal contribution rate. Thus, our findings have good policy reference value for China in efforts to accelerate the development of the digital economy, narrow the regional "digital economic gap" and promote high-quality economic development.

## 2 Theoretical analysis and research hypothesis

This paper discusses the direct impact of the digital economy on high-quality economic development at both the macro and micro levels. From a macro perspective, the digital economy affects economic development by affecting production input and output efficiency, which is reflected in the increase in factor input, the improvement of factor allocation efficiency, and the increase in total factor productivity brought about by technological progress and technological spillover [29]. For developing countries, digitization is considered to be the main driving force of economy growth. It improves capital and labor productivity, reduces transaction costs, and promotes the integration of countries into the global market system [30, 31]. For the developed countries, the impact of the digital economy on the quality of economic development is mainly reflected in promoting sustainable development [32] and improving enterprise agility [8]. The digital economy was the main driving force of economic growth in the United States from 2004 to 2012 [33]. The independent R&D investment and technological progress of its production sector had a significant positive impact on total factor productivity (TFP) growth [34]. Informatization plays a significant role in promoting TFP growth in 12 major countries of the OECD [35]. The data from 2009 to 2018 in the 15 advanced economies of the European Union show that national and industry digital policies can significantly promote

economic growth [8]. In addition, the increase in the productivity of digital assets has brought about a sharp drop in the employment rate in the large-scale production system [36], which also has created a long-term and stabilizing effect on the development of technological innovation [37]. From a micro point of view, digital manufacturing technology improves the competitiveness of enterprises and improves the performance of the company [38], effectively promotes the development of the electronic government [39], and helps enterprises to realize the business model of a circular economy [40]. Generally, despite the rapid development of information technology and digital technology, the digital economy represented by information data and cloud computing has not played its greatest role in economic development [41].

In the era of the digital economy, according to the theory of comparative advantage and first-mover advantage, because of its high technology, high permeability and high growth, for its development, the digital economy depends on platforms, intelligence and ecology to create opportunities for regional economic growth, technological progress and digital infrastructure construction. Therefore, areas with a sound infrastructure and in which the digital economy is developed will have comparative advantages and first-mover advantages, including a high agglomeration of the digital industry, the application of digital technology, digital ecosystems and so on. First-mover advantages include the improvement of the digital infrastructure, human capital accumulation and so on, which will form the driving force for the high-quality development of the regional economy. Based on the above analysis, the research hypothesis H1 is proposed:

H1: The development of the digital economy is conducive to the improvement of total factor productivity, which directly promotes high-quality economic development. Compared with regions with a poor level of digital economy development, regions with a better foundation can benefit from the development of the digital economy.

Total factor productivity refers to the residual value excluding the contribution rate of all tangible production factors in economic growth. Technological progress is an important contribution factor of this residual value, and information technology is a significant attribute of the digital economy. According to the endogenous growth theory, endogenous R&D and innovation are the core elements that drive economic growth and technological progress [42]. The advancement of digital technology is regarded as not only an extension of the production possibility boundary, but also an extension of the innovation possibility boundary [29]. Therefore, the digital economy can improve total factor productivity by promoting technological progress. On the one hand, in the era of the digital economy, the cost structure of the production and operation of enterprises has changed, forming a structure of high fixed cost and low marginal cost. Network externalities gradually enlarge this cost structure, which gradually reduces the average cost of production and operation of enterprises, and produces economies of scale. On the other hand, in the era of a digital economy, enterprises pay more attention to the diversified production of products or services. Enterprises accumulate user data through multilateral platforms and then in the process of developing other products or services, import users of the original platform, reducing operating costs. At the same time, the growth of the digital economy also enables the parallel development of multiple business models to achieve economies of scope. Additionally, information technology can enable the effective integration of the information of both market supply and demand in the same space and period of time on the platform; information technology can be used to improve the matching efficiency of supply and demand and can simplify the information redundancy in the operation of the economic system, reducing market transaction costs and improving the operation efficiency of the economic system. Finally, the development of information technology transforms traditional industries. New generation information technologies, such as big data, cloud computing and

artificial intelligence, are applied to the production, operation, circulation and consumption of traditional industries, with data as the key production factors. The digital division of labour, enable the realization of an industrial ecology of knowledge sharing and factor coordination, and significantly improves the development efficiency of traditional industries. Therefore, this article proposes hypothesis H2:

H2: The development of the digital economy can indirectly have a positive impact on total factor productivity by promoting technological progress.

## 3 The measurement of digital economy in China

### 3.1 The measuring index for interprovincial digital economy in China

The digital economy originated from the fifth technological revolution and its leading technology is information and communications technology. Different from the economies emerging from the previous four technological revolutions, the digital economy presents three typical characteristics. First, data have become the core factors of production and in the supply of the traditional factors of production, resources with data elements as the core are effectively allocated; in addition, the dimension and volume of the supply of data elements is constantly enriched so that data permeates all aspects of economy and social development, reflecting the integration of the development of the digital economy. Second, the digital infrastructure has become a supporting facility, and the new generation of information technology represented by big data, artificial intelligence, mobile Internet and cloud computing has become the leading technology in the development of the digital economy, advancing the information revolution into the era of flexibility, sharing and high-performance computing [43]. In addition, the infrastructure including the communications infrastructure, the Internet of Things terminals, cloud computing resources, etc. has adapted to technological development and entered the fast lane of development. Thirdly, the digital economy has become the main driving force of economic development in this era. With the advantages of having special netizens, China will gradually release its economic growth momentum [44], and its contribution rate to economic growth will increase steadily.

Combined with the above analysis of the essential attributes and characteristics of the digital economy, referring to digital economy reports and white papers issued by authoritative research institutions in China, and following the principles of the timeliness, representativeness, comparability and data availability, this paper holds that the development of the digital economy refers to the use of a new generation of information technology and information infrastructure to infiltrate all aspects of economic and social development; moreover, this development also refers to the undertaking of a series of activities to promote the development of traditional industries and the optimization of the economic structure. Therefore, this paper is constructing an index system to measure the development of China's interprovincial digital economy from three dimensions: the digital infrastructure, the digital industry and the digital fusion effect.

As shown in Table 1, for measuring the development of the digital economy, this paper designs an index system, which includes 3 second-level indicators and 12 third-level indicators, and evaluates the development level of China's interprovincial digital economy. To make the statistical indicators of different regions comparable in the time section, this paper uses the proportional index as much as possible. Additionally, the development effect of digital economy integration is influenced by many factors, which cannot be measured by a single indicator. Therefore, this article draws on the online government index and the digital life index in the China Interprovincial Information Society Index developed by the Ministry of Industry

Table 1. Digital economy measurement index system and its weight.

| First-level inde | Second-level index | Third-level index | Unit | Weight |
|---|---|---|---|---|
| Digital economy development index | Digital economic infrastructure sub-index. (0.4152) | Internet broadband access port | 10000/10000 people | 0.0908 |
| | | Mobile Internet access traffic per capita | 10000 GB/10000 people | 0.0836 |
| | | Mobile phone penetration rate | Per hundred people | 0.1137 |
| | | Number of websites per 100 enterprises | Per 100 enterprises | 0.1271 |
| | Digital economic industrial development sub-index (0.2984) | Value added for information transmission, software and information technology services/GDP | Percentage | 0.0996 |
| | | Value added for computer, communications and other electronic equipment manufacturing/GDP | Percentage | 0.0796 |
| | | Number of enterprises related to digital economy | Number | 0.0645 |
| | | ICT investment level proportion of regional digital investment | Percentage | 0.0547 |
| | Digital economy integration and application sub-index. (0.2864) | Regional e-commerce procurement and sales/regional GDP | Percentage | 0.0399 |
| | | Industrialization information integration index | Third-party data | 0.0903 |
| | | Online government index | Third-party data | 0.0797 |
| | | Digital life index | Third-party data | 0.0765 |

and Information Technology and the National Information Center's China Interprovincial Information Society Index. These three indicators are all structured from second-hand data, and other indicators are based on information from the National Bureau of Statistics and the statistical yearbooks of various provinces and cities. For details of the data standardization and confirmation process, please refer to S1 Appendix.

1. Digital economic infrastructure sub-index. This part of the index is mainly used to measure the development level of the interprovincial digital economic infrastructure, which is an important prerequisite for the regional development of the digital economy. This paper mainly investigates two aspects: the mobile Internet and the Internet ports. The mobile Internet development level is characterized by mobile phone penetration and mobile Internet per capita access traffic. The Internet port development level is explained by the number of Internet bandwidth access ports and enterprise-owned websites.

2. Digital economic industrial development sub-index. This part of the index is mainly used to do the following: measure the industrial scale, enterprises and investment development status of the interprovincial digital economy; explain the core industrial scale of the digital economy with the added value of information technology services and electronic equipment manufacturing industry; explain through the number of enterprises the development level of the regional economy related enterprises and explain through ICT investment the regional digital economic investment.

3. Digital Economy Integration and Application sub-index. This part of the index is mainly used to measure the development level of interprovincial digital economy integration. China's consumer Internet is one of the core components of the digital economy; therefore, the regional e-commerce sales volume is used to measure the development of the regional "digital economy + service industry". The level of industrial digitization cannot be measured by a single index; therefore, this paper draws lessons from the integration development index of the Ministry of Industry and Information Technology to measure it. In addition, the concepts "digital economy + government affairs" and "digital economy + people's livelihood"

are two important aspects of the development of digital fusion. Therefore, to measure the development level of digital economy integration, we draw lessons from the online government index and the digital living index in the information society index issued by the National Information Center.

## 3.2 The present situation of digital economy in China

The changes in China's interprovincial digital economic development index and in its three sub-indices from 2015 to 2019 are shown in Fig 1. During the sample period, the overall development level of China's digital economy showed an upward trend year by year, and the growth trend was relatively stable, rising from 43.39 points in 2015 to 59.31 points in 2019 with an increase of 36.69% and an average annual growth rate of 8.13%. The development level of the digital economy has improved significantly. Regarding the sub-dimensional indices with an average annual growth rate of 12.45%, the digital economic infrastructure sub-index rose from 40.84 in 2015 to 65.29 in 2019, an increase of 59.87%. Since the 13th Five-Year Plan, from the construction of broadband China to the "speed-up and fee reduction" of the telecommunications industry, China has made great efforts to develop its digital infrastructure and to promote China's digital infrastructure construction in an orderly manner and from a global perspective, effectively bridging the regional digital infrastructure gap. The digital economic industrial development sub-index rose from 46.72 in 2015 to 50.39 in 2019, reflecting an increase of 7.87% and an average annual growth rate of only 1.91%. On the whole, the digital industry has not significantly improved the development of the digital economy in China. The reason for this is strongly correlated with China's digital economy's industrial distribution, which is characterized by a concentration of most of the digital core industries in the eastern developed areas. This industrial concentration has led to the lack of resource elements for the development of the digital industry in other central and western regions, which has had a

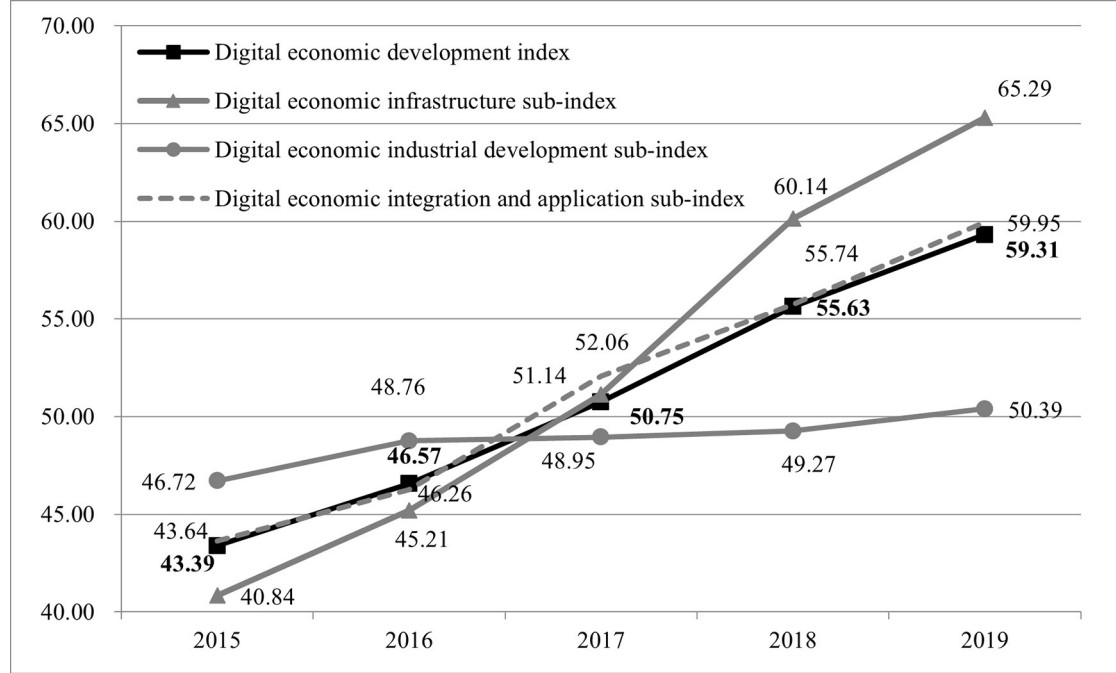

**Fig 1. Changes in China's interprovincial digital economic development index and in its three sub-indices from 2015 to 2019.**

certain impact on the overall development of China's digital industry. The digital economy integration and application sub-index rose from 43.63 in 2015 to 59.95 in 2019, representing an increase of 37.38 percent and an average annual growth rate of 8.26 percent. Since the 13th Five-Year Plan, the state has vigorously promoted the integration of industrialization and information paid attention to the development of e-government, promoted the development of national governance in the direction of digitalization and intelligence and used digital technology in an orderly manner to improve the efficiency of traditional economic and social operations throughout the country.

The contribution in terms of importance to the development of China's digital economy from 2015 to 2019 is in the following order: the digital infrastructure, digital integration and application, and the development of the digital industry, with contribution rates of 163.18%, 101.88% and 21.45%, respectively. From 2015 to 2019, the development of China's digital economy depended more on the construction of the digital infrastructure, and the effect of digital integration also played a certain role in economic and social development. However, the pulling effect of the digital industry on the development of the digital economy has not been shown. One of the possible reasons is the vigorous development of the consumer Internet based on its usage by Chinese netizens, which has also promoted the construction of the digital infrastructure. However, the development of the digital industry represented by the growth of the industrial Internet on the supply side lags behind relatively and the mutually beneficial promotion of industrial development has not yet developed between the regions. Therefore, how to develop the digital economy industry in the future is an important issue on which to focus to promote the development of the digital economy.

## 3.3 The pattern of the interprovincial digital economy in China

As seen from Table 2 below, in the regional pattern of the development of China's digital economy, the eastern region is developed and the non-eastern regions are underdeveloped. In 2019, the average comprehensive index of China's digital economic development was 59.31, and 11 provinces and cities were above the average, including 9 eastern provinces and cities except Hebei as well as Chongqing and Sichuan, while the other 19 provinces and cities were below average. According to the standard of China's National Bureau of Statistics, China's provinces and cities are divided into four regions: the east, the northeast, the central and the west. In the eastern region, with the comprehensive index of the digital economy reaching more than 80 in 2019, Beijing, Guangzhou, Jiangsu and Shanghai are in the first echelon; the second echelon includes Jiangsu and Zhejiang, and the third echelon comprises Tianjin, Hebei, Fujian and Hainan. In the three northeastern provinces, due to the development model of partial industrialization and the lack of digital resource elements, the overall development level of the digital economy is relatively balanced, but the overall development is in the middle and lower levels. In the central region, in the level of development, the digital economy of Anhui and Guizhou is the leader and is more balanced as a whole. In the western region, the provinces and cities represented by Sichuan and Chongqing have a better level of development, while Yunnan, Guangxi, Xinjiang and Gansu have a relatively low level of development.

On the one hand, due to the advantages of a digital infrastructure and a digital resource endowment, the comprehensive index of the digital economy in the developed areas was at a high level in 2015, and there was relatively limited improvement of the index by 2019. In the provinces and cities represented by Guizhou, Inner Mongolia, Chongqing and Guangxi, the increase in the digital economic index was much higher than that of other regions during the sample study period. Guizhou promoted the construction of a big data comprehensive experimental area and vigorously developed the big data industry. Chongqing combined the

**Table 2. Comprehensive index of the interprovincial digital economy in China.**

| Area | | Composite index in 2015 | composite index in 2019 | index movement (%) | Regional average index in 2019 |
|---|---|---|---|---|---|
| Eastern | Beijing | 75.99 | 89.44 | 13.45 | 73.58 |
| | Tianjin | 47.03 | 61.87 | 14.84 | |
| | Hebei | 36.06 | 59.04 | 22.98 | |
| | Shanghai | 73.32 | 83.55 | 10.23 | |
| | Jiangsu | 65.51 | 80.90 | 15.39 | |
| | Zhejiang | 65.79 | 77.65 | 11.86 | |
| | Fujian | 55.73 | 66.15 | 10.42 | |
| | Shandong | 46.43 | 66.20 | 19.77 | |
| | Guangdong | 73.73 | 87.88 | 14.15 | |
| | Hainan | 47.98 | 63.15 | 15.17 | |
| Northeast | Liaoning | 44.46 | 55.81 | 11.35 | 52.18 |
| | Jilin | 34.58 | 52.18 | 17.60 | |
| | Heilongjiang | 33.17 | 48.54 | 15.37 | |
| Central | Shanxi | 33.29 | 46.22 | 12.93 | 50.68 |
| | Jiangxi | 37.57 | 48.88 | 11.31 | |
| | Anhui | 43.04 | 56.05 | 13.01 | |
| | Henan | 34.23 | 49.16 | 14.93 | |
| | Hubei | 40.24 | 55.79 | 15.55 | |
| | Hunan | 37.83 | 47.99 | 10.16 | |
| western | Mongolia | 32.89 | 58.21 | 25.32 | 54.19 |
| | Guangxi | 26.97 | 48.75 | 21.78 | |
| | Chongqing | 41.04 | 63.43 | 22.39 | |
| | Sichuan | 44.95 | 62.27 | 17.32 | |
| | Guizhou | 25.30 | 51.93 | 26.63 | |
| | Yunnan | 28.92 | 40.56 | 11.64 | |
| | Shanxi | 42.36 | 57.15 | 14.79 | |
| | Gansu | 30.05 | 47.06 | 17.01 | |
| | Qinghai | 39.32 | 54.62 | 15.30 | |
| | Ningxia | 35.93 | 56.93 | 21.00 | |
| | Xinjiang | 28.07 | 42.05 | 13.98 | |

advantages of its own traditional industrial manufacturing base and vigorously developed a digital transformation of manufacturing. These areas combined national policy arrangements and their own industrial advantages and resource endowments to improve the development level of the digital economy. In addition, the provinces and cities represented by Yunnan, Liaoning and Jiangxi demonstrated a relatively low increase in the comprehensive index of digital economic development during the sample period, and the development of the digital economy in these underdeveloped areas started relatively late. A backward infrastructure and a lack of human resources has further limited the development of the digital economy, which not only had a certain impact on the current economic and social development but also widened the "digital economic gap" between regions. In the 14th Five-Year Plan, the issue of how to improve the development level of the digital economy both systematically and comprehensively in underdeveloped areas will become an important topic.

On the other hand, from the perspective of China's regional digital economic development, the variances of China's digital economy development index from 2015 to 2019 are 199.14, 184.88, 178.41, 170.40 and 165.58, respectively, which shows that the "digital economic gap" among China's provinces as a whole is constantly narrowing. This narrowing reflects the state's policy support

**Table 3. Classification of China's interprovincial digital economy development level clubs in 2019.**

| Highly developed areas | BeijingGuangdongShanghaiJiangsuZhejiang |
|---|---|
| Moderately developed areas | FujianShandongHainanTianjinChongqingSichuan |
| Low developed areas | Shanxi (western)HebeiLiaoningAnhuiHubeiHenanMongoliaNingxiaQinghai |
| Underdeveloped areas | JilinShanxi (Central)JiangxiHeilongjiangHunanGuangxiGuizhouGansuXinjiangYunnan |

for promoting the balanced regional development of the digital economy in recent years, including strengthening the regional cooperation and exploring the model of regional coordinated development of the digital economy. Among other initiatives, speeding up the construction and promotion of the digital economy demonstration zone, at the same time, has also played a large part in the region's digital economy development based on the regions own resource endowment and industrial structure. In 2019, the variance of the digital economic development index of the eastern, central, northeast and western regions was 119.26, 14.61, 8.81 and 53.30, respectively. Although the development of the digital economy in the eastern region is better, the "digital divide" in the region is significantly larger than that in other regions, which is mainly reflected in the fact that the development level of the digital economy in provinces and cities such as Beijing, Shanghai and Guangzhou are much higher than that in other regions. In the future, how to replicate and spread the digital economy development experience of developed regions to other regions is one of the effective ways to solve the "digital economy gap" among provinces in China.

Table 3 below divides the club classification of the development of China's interprovincial digital economy into four levels, in which the highly developed and moderately developed areas of the digital economy are all above the national average. The other 19 regions, except Hebei are concentrated in the northeast, western and central provinces and cities, reflecting the fact that the development level of China's interprovincial digital economy shows significant regional heterogeneity and an obvious "Matthew effect" [45].

# 4 Model construction

## 4.1 Variable definition

**(1) Dependent variable.** At present, a unified accounting framework and system has not been developed for the academic measurement of high-quality economic development, and the methods can be summarized as multi-index and single-index measurement approaches, in which the multi-index measures are mostly guided by "five major development concepts" that are used to comprehensively and systematically describe the quality of economic development, that is, to describe high-quality economic development in a broad sense. A single index focuses more on the efficiency of economic development in the connotation of high-quality economic development, that is, high-quality economic development in a narrow sense. The purpose of this paper is to study the impact of the digital economy on the efficiency of economic development; therefore, a single index of total factor productivity is used to describe high-quality economic development [46, 47]. This approach also coincides with the view of most scholars that the key to achieving high-quality economic development is to improve economic efficiency. Among the methods of measuring total factor productivity, the Solow residual method is the most common [48]. The Cobb-Douglas production function can be expressed as:

$$Y_{it} = A_{it}K_{it}^{a}L_{it}^{1-a} \; 0 < a < 1, \; A_{it} = {Y_{it}} \big/ {K_{it}^{a}L_{it}^{1-a}} \tag{1}$$

**Table 4. Variable definition.**

| Variable | Symbol | Basic meaning | Measure |
|---|---|---|---|
| Dependent variable | TFP | Total factor production efficiency | The C-D function is used to measure Inter-provincial Total Factor Productivity |
| Independent variable | DEI | Digital economy development index | The digital economy index consists of three dimensions: the basic sub-index (INF), the industrial sub-index (DIND) and the integration sub-index (FUSE). |
| Intermediary variable | TP | Technological progress | Combination of regional patent application authorization and technology market turnover |
| Control variable | TD | Level of technology development | R&D investment of industrial enterprises/ regional GDP |
| | FD | Level of financial development | Total deposits and the loan balance of financial institutions/regional GDP |
| | IS | Level of industrial structure | Service industry value added/regional GDP |
| | OPEN | Level of opening up | Total imports and exports/regional GDP |
| | RD | Level of financial R&D investment | Government financial R&D expenditure/general budget expenditure of local finance |

Among them, capital investment K is measured by using the internationally accepted perpetual inventory method to estimate the physical capital stock of each province and city in the sample period. The specific formula is $K_t = I_t + (1-\delta)K_{t-1}$, and the estimation of the physical capital stock at the beginning of the period is $K_0 = I_0/(g+\delta)$. Taking into account that the base period is set earlier, the initial capital stock estimation will have less impact on the later capital stock estimation. This article sets the base period in 2010. $I_t$ is the investment amount in year $t$, expressed by the total fixed asset investment of the whole society, and is converted to 2010 constant prices RMB by the price index; $\delta$ is the fixed asset depreciation rate. This article refers to the practice of most scholars to calculate the capital stock of all provinces and cities across the country and sets $\delta = 9.6\%$. In the calculation of total factor productivity, different documents have certain differences in the calculation of the value of capital output elasticity $a$. In the process of calculating the total factor productivity of various provinces and cities, this paper studies the value of capital output elasticity $a$ as 0.6 [49]. The statistical description of the measured data is shown in Table 3 (TFP). In addition, the capital output elasticity $a$ is set to 0.5 and 0.7 for the robustness test.

**(2) Independent variable.** The digital economic index of the core independent variable is calculated by a total of 12 indicators from three dimensions: the infrastructure sub-index, the digital industry sub-index and the digital fusion sub-index. The indicators that reflect the level of technological progress among provinces in China are calculated comprehensively by two indicators, namely, regional patent application authorizations and regional technology market turnover, and the data come from the National Bureau of Statistics. Since the process of index standardization and weighting is similar to the process of measuring digital economic index, we will not repeat it here.

The definitions of relevant variables in this paper are shown in Table 4, and the descriptive statistics of the variables are shown in Table 5.

## 4.2 Research methodology

**(1) Direct effects model.** Construct a multiple linear regression model to explore the direct effect of the digital economy on high-quality economic development (total factor productivity). The model form is as follows:

$$TFP_{it} = \beta_0 + \beta_1 DEI_{it} + Controls + \mu_{it} \qquad (2)$$

**Table 5. Statistical description of standardized data.**

| Variable | Sample size | Mean value | Standard deviation | Minimum value | Maximum value |
|----------|-------------|------------|---------------------|----------------|----------------|
| TFP | 150 | 52.4840 | 11.2899 | 34.8840 | 77.6635 |
| DEI | 150 | 51.1314 | 14.6046 | 25.2996 | 89.4383 |
| TP | 150 | 49.9420 | 14.2785 | 36.2121 | 100.3963 |
| TD | 150 | 47.9432 | 28.7876 | 5.8856 | 99.9783 |
| FD | 150 | 48.0834 | 28.3672 | 3.2096 | 99.2884 |
| IS | 150 | 45.3988 | 21.5884 | 8.7335 | 100.0000 |
| OPEN | 150 | 45.3853 | 20.8818 | 23.9121 | 99.4366 |
| RD | 150 | 49.3396 | 22.2278 | 24.2445 | 108.8159 |

In model (2), $TFP_{it}$ represents the province's total factor productivity, $DEI_{it}$ represents the level of digital economy development, and the indicator data come from the previous measurements of the digital economy index. According to the previous analysis, the development of the digital economy can increase total factor productivity by promoting technological progress. Regarding control variables, consistent with existing research practices [50], technological progress (TP), level of technology development (TD), level of financial development (FD), level of industrial structure (IS), level of opening up (OPEN) and level of financial R&D investment (RD) are added. $i$ represents the province, $t$ represents the year, $\beta_0$ represents the intercept term, and $\mu_{it}$ represents the random error term.

Using the general benchmark model, this paper tests the impact of the three dimensions of the digital economic index, namely, the basic sub-index (INF), the industrial sub-index (DIND) and the integration sub-index (FUSE), on total factor productivity. The model setting is similar to model (2).

**(2) Mediating effect model.** This paper introduces the mediation effect model to study whether the digital economy can increase the total factor productivity by promoting technological progress when technological progress is used as an intermediate variable, thereby promoting the high-quality development of China's economy. The specific mediation effect model is set as follows:

$$TFP_{it} = C + \alpha DEI_{it} + Controls + \mu_{it} \tag{3}$$

Compared with those in model (2), the controls in this model do not include technical progress indicators, and the definition of other variables is the same as that in model (2).

The second step is to verify the impact of the digital economy on promoting technological progress. First, we take technological progress as the dependent variable and the digital economy index as the independent variable to test the effect of the digital economy on technological progress, and we establish a panel model:

$$TP_{it} = C + \eta DEI_{it} + Controls + \mu_{it} \tag{4}$$

The third step is to test whether the mediating effect of technological progress is complete, that is, whether the digital economy can directly increase total factor productivity. This paper constructs the following panel model:

$$TFP_{it} = C + \theta DEI_{it} + \lambda TP_{it} + Controls + \mu_{it} \tag{5}$$

In the above model, technological progress is the mediating variable. The coefficient $\alpha$ in model (3) is the total effect of the digital economy in improving total factor productivity, the coefficient $\eta$ in model (4) is the effect of the digital economy on the intermediary variables,

and the coefficient $\lambda$ in model (5) controls for the development of the digital economy After the impact of the digital economy on the intermediary variables is determined, we examine the effect of the intermediary variables on total factor productivity; $\theta$ is the direct effect of the digital economy on improving total factor productivity, and the mediating effect is $\eta\lambda$, which has the following relationship with the total effect $\alpha$, and the direct effect $\theta$:

$$\alpha = \theta + \eta\lambda \qquad (6)$$

Regarding whether there is a mediating effect and the extent of the mediating effect, the test process of the mediating effect is as follows: if the coefficient $\alpha$ is significant and both $\eta$ and $\lambda$ are significant, the mediating effect is significant; if the coefficient $\alpha$ is not significant or both $\eta$ and $\lambda$ are not significant, there is no mediating effect. If the coefficient $\alpha$ is significant, both $\eta$ and $\lambda$ are significant, and the coefficient $\theta$ is less than the coefficient $\alpha$, then technological progress is part of the mediation variable, in which case, the mediation effect accounts for $\eta\lambda$ / $\alpha$. If the coefficient $\alpha$ is significant, both $\eta$ and $\lambda$ are significant, but $\theta$ is not significant, then there is a complete mediation effect; that is, the effect of the digital economy in improving total factor productivity must be fully exerted through the mediation effect.

## 5 Results and discussion

### 5.1 Direct effect test

This paper uses the panel data from 30 provinces and cities in China from 2015 to 2019 for estimation. Since both the LR test and Wald test show that the random disturbance term in the total benchmark model has heteroscedasticity between groups, the Wooldridge test shows that the disturbance term has the first order within the group. Relatedly, this article adopts feasible generalized least squares (FGLS) to estimate the parameters to overcome the influence of heteroscedasticity and autocorrelation and adopts the OLS and GEE (generalized estimating equation) methods to test the robustness. In the OLS model estimation, the time effect and individual effect of the regional sample are controlled. The OLS model estimation in the following is the same. The regression results of the general benchmark regression model and each sub-dimension index are shown in Table 6 (This paper uses Excel software for data statistics and Stata 15.0 software for regression analysis).

As shown in Table 6, column (1) shows the FGLS estimation result of the general benchmark model. The digital economy has a significant positive effect on total factor productivity, with an impact coefficient of 0.4551, and is significant at the level of 1%, which indicates that in China's digital economy, every one percentage point increase in the level of development can directly increase the total factor productivity by 0.4551 percentage points, which proves that the digital economy plays an important role in increasing the total factor productivity. In another core variable, the influence coefficient of technological progress on the improvement of total factor productivity reaches 0.1527, which is also significant at the level of 1%, which can indicate that technological progress has a certain effect on improving China's interprovincial total factor productivity but its influence coefficient is obviously low. This shows that the development of the digital economy and the improvement of the level of technological progress both have a certain positive impact on the improvement of regional total factor productivity, but the impact of the digital economy is significantly greater than that of technological progress. This shows that the development of the digital economy generates more advantages than just the advantage of raising the level of technological progress and has a deeper and broader connotation and meaning. Regarding the other control variables, the coefficients of influence on total factor productivity in descending order are opening up, financial development, technology development, industrial structure and financial subsidies. The influence

**Table 6. The regression results of the digital economy index and its infrastructure sub-index, industry sub-index, and integration sub-index.**

| variable | Dependent variable: total factor productivity (TFP) | | | | | |
|---|---|---|---|---|---|---|
| | (1) | (2) | (3) | (4) | (5) | (6) |
| Estimation method | FGLS | | | | GEE | OLS |
| DEI | 0.4551***(10.88) | | | | 0.2933***(12.64) | 0.4551***(10.58) |
| INF | | 0.2452***(10.13) | | | | |
| DIND | | | 0.0773**(1.23) | | | |
| FUSE | | | | 0.3458***(7.88) | | |
| TP | 0.1527*** | 0.2306*** | 0.3678*** | 0.2377*** | 0.3132*** | 0.1527*** |
| | (3.55) | (5.60) | (6.68) | (5.17) | (8.20) | (3.45) |
| TD | 0.0456*** | 0.0553*** | 0.0492** | 0.01857** | 0.0159* | 0.0456** |
| | (3.11) | (3.65) | (2.51) | (1.10) | (1.04) | (3.02) |
| FD | 0.0565*** | 0.0712*** | 0.0504** | 0.0129* | 0.0226** | 0.0565*** |
| | (3.34) | (4.05) | (2.24) | (0.66) | (2.03) | (3.25) |
| IS | 0.0379** | -0.0359** | 0.0454** | -0.0249** | 0.0276* | 0.0379** |
| | (-2.20) | (-2.02) | (2.18) | (-1.28) | (1.73) | (2.14) |
| OPEN | 0.1195*** | 0.1909*** | 0.1976*** | 0.1619*** | 0.0668** | 0.1195*** |
| | (3.78) | (6.14) | (4.18) | (4.66) | (2.44) | (3.68) |
| RD | -0.0910*** | -0.0986*** | -0.1067*** | -0.0791*** | -0.0634*** | -0.0910*** |
| | (-3.69) | (-3.88) | (-3.26) | (-2.83) | (-2.68) | (-3.59) |
| constant | 17.4683*** | 19.8441*** | 19.7913*** | 18.9658*** | 21.3497*** | 17.4683*** |
| | (15.13) | (17.26) | (11.51) | (14.89) | (12.50) | (14.72) |
| Wald | 1748.93 | 1638.11 | 922.44 | 1351.01 | 914.74 | - |
| $R^2$ | - | - | - | - | - | 0.9171 |
| N | 150 | 150 | 150 | 150 | 150 | 150 |

Note: The Z statistic values are in parentheses, and

***, **, * indicate significance at the levels of 1%, 5%, and 10%, respectively. The same applies to the following table.

coefficient of financial subsidies on total factor productivity is significantly negative. Appropriate government financial subsidies can improve the performance of enterprises, thereby improving the operating efficiency of the entire economic system, while excessive subsidies have a negative impact on corporate performance [51]. In addition, incidents of collusion between local governments and enterprises are common in China and fiscal subsidies based on political connections can distort the effective allocation of scarce resources in the entire society [52]. The empirical data in this article show that at this stage, fiscal subsidies have not produced good effects and have even produced certain side effects. Therefore, as subsidies are currently not efficient, it is important to understand how to combine the information technology characteristics of the digital economy to change the previous extensive and rent-seeking financial subsidy model and to provide "precise subsidies" to enterprises with high R&D efficiency to achieve "making full use of money". In addition, columns (5) and (6) display the GEE and OLS regression results of the general benchmark model, respectively. The results show that the digital economy and technological progress play a significant role in improving total factor productivity and both are significant at the 0.01 level. The significance of other control variables is basically consistent with the result of FGLS estimation, which shows that the FGLS estimation of the general benchmark model in this paper has good stability.

The regression results of the effects of the infrastructure sub-index, industrial sub-index and integration sub-index on total factor productivity are shown in columns (2) to (4) of Table 6. The influence coefficients are 0.2452, 0.0773 and 0.3458. The industrial sub-index is

significant at the 5% level, and the infrastructure sub-index and integration sub-index are significant at the 1% level. The influence coefficients ranked from large to small are the integration sub-index, the infrastructure sub-index, and the industry sub-index. Combining the influence coefficient of each sub-index reveals that at present, the role of China's digital economy in improving total factor productivity is mainly reflected in the integration effect, mainly demonstrated in the rapid development of China's consumer Internet and e-commerce, and the supporting information infrastructure has also been improved. However, as the core component of the digital economy, the role of the digital industry in promoting total factor productivity is not as good as that of the former two, and the development of basic innovation and information technology depends on the high-quality development of the digital industry. Compared with the digital infrastructure and digital integration and application, the digital industry is a driving force for the development of the digital economy. Therefore, for areas with a good digital infrastructure, it is necessary to focus on the development of the digital industry to realize the transformation of the industrial structure and improve total factor productivity in the future.

## 5.2 Mediating effect test

The above analysis confirms that the digital economic index and its fractal index play a significant role in improving total factor productivity. Consistent with the previous theoretical analysis, technological progress is an important basis for giving full play to the role of the digital economy in improving total factor productivity, and the continuous improvement of the development level of the digital economy has also had a certain impact on technological progress. Therefore, it is necessary to identify the internal mechanism of the development of the digital economy in improving total factor productivity from the perspective of technological progress. On the one hand, the digital economy is a new form of economic development that occurs after the era of the industrial economy and that is different from the previous economic development model. The digital economy has significant technical effects, scale effects and multilateral effects. Through the use of intelligent technology to improve the previous production, manufacturing, circulation, transaction and other links, compared with previous economic forms, the digital economy generates higher production efficiency. On the other hand, the digital economy is a technology-economic paradigm [53]. The rapid development of information technology lays a good foundation for the digital economy. The development of a new generation of information technology represented by big data, cloud computing, the Internet of Things and artificial intelligence is a strong support mechanism that gives full play to the efficiency of the digital economy. Therefore, does technological progress play a significant role in the process of facilitating the ability of the digital economy to improve total factor productivity? What is the degree of contribution? Next, this paper uses the stepwise regression estimation method to carry out an in-depth study of these problems.

In testing the comprehensive effect of the digital economy on the improvement of total factor productivity, the intermediary variables of technological progress are excluded. Table 7, columns (1)~(3) shows the results of the FGLS, GEE and OLS estimation of the effect of the digital economy on total factor productivity. The elasticity coefficients of the digital economy to the improvement of total factor productivity based on the regression results of FGLS, GEE and OLS are 0.5335, 0.3548 and 0.5133, respectively, which are all significant at the level of 0.01. Moreover, in the estimation of the FGLS method for eliminating heteroscedasticity and autocorrelation among variables, the regression coefficient is higher than that of the other two estimation methods, indicating that endogeneity problems will underestimate the comprehensive effects. Therefore, this paper mainly uses FGLS as the main estimation method. Because

**Table 7. The comprehensive effect of the digital economy on the improvement of total factor productivity and the intermediary effect of technological progress.**

| Variable | Comprehensive effect: Dependent variable: total factor productivity (TFP) | | | Mediating effect Dependent variable: technological progress (TP) | | |
|---|---|---|---|---|---|---|
| | (1) | (2) | (3) | (4) | (5) | (6) |
| Estimation method | FGLS | GEE | OLS | FGLS | GEE | OLS |
| DEI | 0.5335*** | 0.3548*** | 0.5335*** | 0.5133*** | 0.1676*** | 0.5133*** |
| | (14.41) | (13.11) | (14.07) | (7.61) | (3.49) | (7.43) |
| TD | 0.0542*** | 0.0120* | 0.0542*** | 0.0563** | -0.0315** | 0.0563** |
| | (3.59) | (0.66) | (3.51) | (2.05) | (-0.97) | (2.00) |
| FD | 0.0611*** | 0.0262** | 0.0611*** | 0.0307* | 0.0038* | 0.0307** |
| | (3.48) | (1.92) | (3.40) | (0.96) | (0.16) | (0.94) |
| IS | 0.0583*** | 0.0011* | 0.05827*** | 0.1336*** | 0.1089*** | 0.1336*** |
| | (2.24) | (0.06) | (3.37) | (4.35) | (3.34) | (4.25) |
| OPEN | 0.1238*** | 0.1189*** | 0.1238*** | 0.0280* | 0.1236** | 0.0280* |
| | (3.77) | (3.78) | (3.68) | (0.47) | (2.17) | (0.46) |
| RD | -0.0555** | 0.02479** | -0.0555** | 0.2328** | 0.2889*** | 0.2328*** |
| | (-2.37) | (0.97) | (-2.31) | (5.44) | (6.42) | (5.31) |
| constant | 19.4279*** | 25.5392*** | 19.4279*** | 12.8311*** | 17.8944*** | 12.8311*** |
| | (18.40) | (12.48) | (17.96) | (6.66) | (5.56) | (6.51) |
| Wald | 1601.69 | 600.74 | - | 692.87 | 280.79 | - |
| $R^2$ | - | - | 0.9108 | - | - | 0.8146 |
| N | 150 | 150 | 150 | 150 | 150 | 150 |

Note: The Z statistic values are in parentheses, and

***, **, * indicate significance at the levels of 1%, 5%, and 10%, respectively. The same applies to the following table.

the influence coefficient of the digital economic index in the comprehensive effect is significantly positive, which accords with the preliminary test step of the intermediary effect, this paper argues according to the existence of the intermediary effect, that is, the assertion that the comprehensive effect includes direct effect and indirect effect, and then analyses the intermediary effect of technological progress.

Table 7, columns (4)~(6) calculate the effect of the digital economy on the level of regional technological progress. The regression coefficients of FGLS, GEE and OLS are 0.5133, 0.1676 and 0.5335, respectively, and they all pass the significance test of 0.01. It can be seen that the influence of the development of the digital economy on the level of technological progress in the region is robust, and the results show that the development of the digital economy has a significant positive effect on improving the level of technological progress in the region.

In order to examine the influence mechanism by which the digital economy improves total factor productivity, to test whether technological progress is a complete intermediary variable and whether there are direct effects in its influence process, model (4) brings the direct and indirect effects of digital economic development into the same model for analysis and still uses FGLS, GEE and OLS methods to estimate parameters. As shown in the results of (1), (5) and (6) of Table 6, the estimated coefficients of technological progress are 0.1527, 0.3132 and 0.1527, respectively, and the regression coefficient is still significant. It can be concluded that the intermediary effect of technological progress is significant. The results of Table 7 (1) show that the total effect of the digital economy on the improvement of total factor productivity is 0.5335, and those of Table 7 (4) show that the effect of the digital economy on the improvement of regional technological progress is 0.5133. The proportion of the intermediary effect of

technological progress in the total effect is 14.69%, which shows that technological progress has a certain transmission effect between digital economy and total factor productivity.

## 5.3 Regional heterogeneity analysis

Existing academic studies and related reports have pointed out that the development level of the digital economy in the eastern region is significantly higher than that in the central and western regions, but there is no practical explanation for the impact of regional heterogeneity on high-quality economic development. Therefore, to study whether there is a significant difference in the effect of the digital economy on high-quality economic development between the eastern region and the non-eastern regions, this paper first adds the interprovincial digital economic development index and the interaction term $DEI \times dum$ between the eastern region and the non-eastern region through the use of formula (6), in which $dum = 1$ represents the eastern region and $dum = 0$ represents the non-eastern region (according to China's National Bureau of Statistics, the eastern region includes 10 provinces and cities, including Beijing, Tianjin, Hebei, Shanghai, Jiangsu, Zhejiang, Fujian, Shandong, Guangdong and Hainan, while the rest of the northeast, central and western regions are divided into the non-eastern regions), and the parameters are estimated.

$$TFP_{it} = \beta_0 + \beta_1 DEI_{it} + \beta_2 DEI_{it} \times dum + Controls + \mu_{it} \tag{7}$$

In addition to studying the effect of the comprehensive index of the digital economy, this article will continue to study the interaction effects between the three sub-dimensions of the digital infrastructure, the digital industry, and digital integration and the regions. The empirical model setting is similar to formula (6) and will not be repeated here.

The results of (1), (5) and (6) of Table 8 estimate the interaction terms between the digital economy index and the regions by the FGLS, GEE and OLS methods, and the influence coefficients are 0.0968, 0.0619 and 0.0968, respectively; the results are all significant at the level of 1%. This result has good robustness, indicating that the effect of the digital economy on the promotion of total factor productivity is more significant in the eastern region. Due to the early start of the development of the digital economy in the eastern region and because the digital infrastructure and resource elements in the eastern region are more perfect than those in the non-eastern regions, the eastern region has a certain first-mover advantage and comparative advantage. Because the non-eastern regions lag behind the eastern region in the stage of economic development, the lag of the development of the digital economy further limits its positive effect on the improvement of total factor productivity. According to the three sub-indices of the digital economy index, the regression results of the infrastructure sub-index, the industrial sub-index, the integration sub-index with the regional interaction items are 0.0677, -0.4376 and -0.2336, respectively. This paper uses the "digital access gap", "digital industry gap" and "digital application gap" to represent these three interactive items. The influence coefficient of the digital access gap is the largest, that of the digital application gap is the second largest, and that of the digital industry gap is the smallest. On the one hand, there is no strict regional boundary for digital applications in China. Due to Internet platform products or services available throughout the country, enterprises and individuals in all regions can access digital application products or services. The "digital divide" is more reflected in the "digital access gap", which once again verifies that the eastern region, due to its more perfect digital infrastructure, has greater advantages in the development of the digital economy in improving total factor productivity. According to the 45th Statistical report on the Development of Internet Networks in China, regarding download speeds in the eastern, central and western regions, the average download rate of 4G mobile broadband users in the eastern region reached 24.60

**Table 8. Regression results of the digital economy index and its three sub-indexes and regional interaction items.**

| Variable | Dependent variable: total factor productivity (TFP) | | | | | |
|---|---|---|---|---|---|---|
| | (1) | (2) | (3) | (4) | (5) | (6) |
| Estimation method | FGLS | | | | GEE | OLS |
| DEI | 0.4609*** | | | | 0.2706*** | 0.4609*** |
| | (10.29) | | | | (11.98) | (9.95) |
| INF | | 0.2215*** | | | | |
| | | (8.13) | | | | |
| DIND | | | 0.2251*** | | | |
| | | | (3.37) | | | |
| FUSE | | | | 0.3744*** | | |
| | | | | (8.51) | | |
| DEI*DUM | 0.0968*** | | | | 0.0619*** | 0.0968*** |
| | (1.27) | | | | (1.90) | (1.22) |
| INF*DUM | | 0.0677*** | | | | |
| | | (1.36) | | | | |
| DIND*DUM | | | -0.4376*** | | | |
| | | | (-4.75) | | | |
| FUSE*DUM | | | | -0.2336*** | | |
| | | | | (-3.64) | | |
| TP | 0.1815*** | 0.0631*** | 0.4583*** | 0.2748*** | 0.2647*** | 0.1815*** |
| | (3.85) | (3.49) | (8.53) | (6.16) | (6.29) | (3.72) |
| TD | 0.0305** | 0.0688*** | 0.0135* | 0.0034* | 0.0118** | 0.0305** |
| | (1.75) | (3.96) | (0.71) | (0.20) | (0.79) | (1.69) |
| FD | 0.0557*** | 0.0600*** | 0.0389*** | 0.0457*** | 0.01407** | 0.0562*** |
| | (3.30) | (3.47) | (1.88) | (2.56) | (1.32) | (3.23) |
| IS | 0.0377** | 0.1578** | 0.0246** | 0.0416** | 0.0177** | 0.0368** |
| | (2.20) | (4.49) | (1.28) | (2.37) | (1.16) | (2.09) |
| OPEN | 0.1060*** | 0.1230*** | 0.2692*** | 0.1924*** | 0.0093* | 0.1305*** |
| | (3.13) | (3.07) | (5.18) | (4.90) | (0.31) | (3.25) |
| RD | -0.0852*** | -0.0941*** | -0.0905*** | -0.0600*** | -0.0566** | -0.0823*** |
| | (-3.39) | (-3.65) | (-2.96) | (-2.21) | (-2.50) | (-3.17) |
| constant | 15.6321*** | 22.2824*** | 8.5929** | 14.4529*** | 24.8301*** | 15.6321*** |
| | (6.78) | (12.52) | (2.67) | (7.15) | (12.99) | (6.55) |
| Wald | 1784.56 | 1683.83 | 1144.91 | 1521.54 | 1037.92 | - |
| $R^2$ | - | - | - | - | - | 0.9175 |
| N | 150 | 150 | 150 | 150 | 150 | 150 |

Note: The Z statistic values are in parentheses, and

***, **, * indicate significance at the levels of 1%, 5%, and 10%, respectively. The same applies to the following table.

megabytes at the end of 2019, while the central region and the western region were lower by 0.93 Mbit/s and 1.58 megabytes, respectively, showing an obvious digital infrastructure gap. On the other hand, there is also a significant difference in the impact of the "digital industry gap" on the promotion of total factor productivity between the eastern and non-eastern regions. Regarding the progress of industrial development, the eastern region has always been at the forefront and ahead of the non-eastern regions, and digital transformation is the only way to build a modern industrial system. According to the 2018 report on the Digital Development of Chinese Enterprises released by the International Data Company (IDC), China's retail,

**Table 9. Regression results of the regional digital economy in the improvement of total factor productivity.**

| Variable | Dependent variable: total factor productivity (TFP) | | | |
|---|---|---|---|---|
| | Eastern China | Northeast China | Central China | Western China |
| Estimation method | FGLS | | | |
| DEI | 0.6170*** | 0.3398*** | 0.1267*** | 0.3688*** |
| | (8.49) | (4.40) | (1.21) | (6.70) |
| TP | 0.0193* | 0.1272** | 0.8072*** | 0.2621** |
| | (0.29) | (0.67) | (8.43) | (3.49) |
| TD | 0.0115** | 0.0140 | 0.0149* | 0.0677** |
| | (0.53) | (0.41) | (0.51) | (1.71) |
| FD | 0.0574*** | -0.0433*** | 0.0492** | 0.06216** |
| | (2.63) | (-3.07) | (1.96) | (1.86) |
| IS | 0.0594*** | 0.0203* | 0.0158 | 0.0553* |
| | (2.57) | (0.59) | (0.45) | (1.61) |
| OPEN | -0.0458 | 0.1923** | 0.3748*** | 0.3477*** |
| | (-0.66) | (1.50) | (2.58) | (3.50) |
| RD | 0.0676 | 0.1009* | -0.1708*** | -0.0800* |
| | (0.79) | (0.95) | (-5.70) | (-1.32) |
| constant | 21.9107*** | 15.3257*** | 7.4225*** | 8.4486*** |
| | (8.37) | (2.92) | (5.08) | (1.94) |
| Wald | 734.95 | 733.26 | 308.85 | 354.72 |

Note: The Z statistic values are in parentheses, and

***, **, * indicate significance at the levels of 1%, 5%, and 10%, respectively. The same applies to the following table.

entertainment, financial and other consumer enterprises have experienced a high degree of digital transformation. While most of these enterprises are concentrated in the eastern region, the degree of digitization of manufacturing and resource industries is relatively low, and most of the manufacturing industries with successful digital transformation are concentrated in the eastern region. On the other hand, the enterprises with digital transformation demonstrated by individual cases and in some local areas are mostly concentrated in the non-eastern regions, resulting in a significant "digital industry gap" between the eastern and non-eastern regions.

The above analysis shows that in eastern China, the impact of digital economic development on the improvement of total factor productivity is more significant than that in non-eastern regions. In view of the great differences in economic development and digital economy development in different regions of China, this paper divides the samples into eastern, northeastern, central and western regions to examine the impact of digital economy development in different regions on the improvement of total factor productivity. In addition, technological progress may play a more significant intermediary role in different regions.

As shown in Table 9, the influence coefficients of the digital economy index on total factor productivity in the eastern, northeastern, central and western regions are 0.6170, 0.3398, 0.1267 and 0.3688, respectively, and they are all significant at the 1% level. These results show that the development of the digital economy has a significant impact on the improvement of total factor productivity, and it also proves the robustness of the previous results. In addition, the influence coefficient of the digital economic index in the eastern region is significantly larger than that in the other three regions, which also proves that the better level of economic development in the eastern region is an important factor in the higher level of digital economic development in that region. The influence coefficients of technological progress on total factor productivity in the eastern, northeast, central and western regions are 0.0193, 0.1272, 0.8072

and 0.2621, respectively. The influence coefficient of technological progress on total factor productivity in the central region is slightly higher than that in the other three regions. One of the possible explanations is that the marginal impact of technological progress on the improvement of total factor productivity in the central region was greater than that in the other regions during the sample period. The reason and mechanism of this need to be further studied.

As shown in Table 10, the influence coefficients of the digital economy index on technological progress in the eastern, northeast, central and western regions are 0.5425, 0.2472, 0.4531 and 0.2986, respectively, and they are all significant at the 1% level. These results show that the development of the digital economy has a significant impact on improving the level of technological progress, but the influence coefficient in the eastern region is significantly greater than that in other regions. This shows that the benign mutual promotion mechanism between digital economic development and technological progress in the eastern region is better than that in the other three regions. In addition, in the eastern, northeastern, central and western regions, the intermediary effects of technological progress in the influence mechanism promoting the digital economy to improve total factor productivity are 1.70%, 9.25%, 28.89% and 21.22%, respectively. However, at the overall level of China, the intermediary effect of technological progress in promoting the improvement of total factor productivity in the digital economy is 31.31%. It can be seen that the mediating effect of technological progress is more significant in the central and western regions. From another perspective, it shows that the central and western regions, which are at the middle level of digital economy development, are more important to improve the level of technological progress in the process of achieving high-quality economic development through the development of digital economy. In the future, how to effectively deal with the imbalance of regional economic development aggravated by the "digital economic gap" is the issue that must be addressed to realize the balanced development of China's economy.

**Table 10. The regression results of the sub-regional digital economy in promoting technological progress.**

| Variable | Dependent variable: technological progress (TP) | | | |
|---|---|---|---|---|
| | Eastern China | Northeast China | Central China | Western China |
| Estimation method | FGLS | | | |
| DEI | 0.5425*** | 0.2472*** | 0.4531*** | 0.2986*** |
| | (6.70) | (2.94) | (4.08) | (3.31) |
| TD | 0.0022 | 0.0385** | 0.0177* | 0.0287** |
| | (0.05) | (0.84) | (0.32) | (0.41) |
| FD | 0.0101* | 0.0111 | 0.1100*** | 0.1309** |
| | (0.22) | (0.58) | (2.53) | (2.29) |
| IS | 0.2227*** | 0.0358* | 0.1262** | 0.0056 |
| | (6.00) | (0.78) | (2.01) | (0.09) |
| OPEN | 0.6189*** | 0.3902** | 0.5630** | 0.1433 |
| | (5.33) | (2.72) | (2.19) | (0.81) |
| RD | 0.9006*** | 0.2044* | 0.1629*** | 0.1051** |
| | (7.02) | (1.51) | (3.34) | (0.98) |
| constant | 6.8760*** | 12.8082*** | 27.6913*** | 17.7537*** |
| | (1.27) | (2.01) | (3.07) | (2.39) |
| Wald | 564.90 | 204.59 | 192.34 | 342.15 |

Note: The Z statistic values are in parentheses, and

***, **, * indicate significance at the levels of 1%, 5%, and 10%, respectively. The same applies to the following table.

## 5.4 Robustness test

To verify the stability of the results, this paper assumes that the core parameters of total factor productivity are $a = 0.5$ and $a = 0.7$; the entropy method and equality method are used to calculate the digital economic development index, and the robustness test from the above four aspects is conducted. Table 11 shows the results of the four tests, and the regression results are basically consistent with the estimated results of the previous general benchmark model, indicating that the conclusions of this paper are robust.

## 5.5 Endogenous test

The benchmark regression results initially verified the promotion effect of the digital economy on total factor productivity, but there may be endogenous problems in the process of model causality identification. One source of endogeneity is reverse causality, that is, while the development of the digital economy promotes the high-quality development of the regional economy, the high-quality development of the regional economy is in turn promoting the development of the digital economy. In response to this problem, this article refers to the research [54], and uses the one-period lag of DEI as an independent variable for regression. Three methods of FGLS, GEE, and OLS were used to endogeneity testing. And the regression results are shown in Table 12. The influence coefficients of the digital economy index on total factor productivity are 0.4499, 0.3048, and 0.4499 respectively, and they are all significant at the 1% level. This result is very robust. The regression results show that after considering the possible endogenous problems of the model, the development of the digital economy still has a significant positive role in promoting total factor productivity, which further supports the

**Table 11. Robustness test results.**

| Variable | Dependent variable: total factor productivity (TFP) | | | |
|---|---|---|---|---|
| | elasticity of capital output $a$ is 0.5 | elasticity of capital output $a$ is 0.7 | Entropy method to determine the weight | Equality method to determine the weight |
| Estimation method | FGLS | | | |
| DEI | 0.3029*** | 0.4569*** | 0.4267*** | 0.3279*** |
| | (3.89) | (5.68) | (3.12) | (4.05) |
| TP | 0.2128*** | 0.3036*** | 0.1409*** | 0.2026*** |
| | (2.88) | (5.51) | (3.18) | (5.27) |
| TD | 0.0456*** | 0.0688*** | 0.1022*** | 0.0855*** |
| | (1.88) | (3.96) | (3.24) | (2.24) |
| FD | 0.0519*** | 0.0692*** | 0.0456*** | 0.0622*** |
| | (1.95) | (3.37) | (1.24) | (3.25) |
| IS | 0.0218** | 0.0536** | 0.0749** | 0.0618** |
| | (2.21) | (2.97) | (2.93) | (1.15) |
| OPEN | 0.0921** | 0.1519** | 0.0824** | 0.1298** |
| | (2.14) | (3.54) | (1.96) | (2.07) |
| RD | -0.0732** | -0.1926** | -0.0967*** | -0.0876*** |
| | (2.64) | (-1.58) | (-2.58) | (-2.69) |
| constant | 17.9625*** | 19.6258*** | 15.0638*** | 12.3567*** |
| | (6.25) | (12.02) | (11.58) | (9.70) |
| Wald | 1253.24 | 1389.07 | 1104.25 | 956.12 |
| N | 150 | 150 | 150 | 150 |

**Table 12. Endogenous test results.**

| Variable | Comprehensive effect: Dependent variable: total factor productivity (TFP) | | |
|---|---|---|---|
| | (1) | (2) | (3) |
| Estimation method | FGLS | GEE | OLS |
| DEI | 0.4499*** | 0.3048*** | 0.4499*** |
| | (8.47) | (12.42) | (8.18) |
| TD | 0.1547*** | 0.2193*** | 0.1547*** |
| | (3.23) | (5.62) | (3.12) |
| FD | 0.0465*** | 0.0208** | 0.0465*** |
| | (2.81) | (1.43) | (2.71) |
| IS | 0.0562*** | 0.0040* | 0.0562*** |
| | (2.89) | (0.40) | (2.79) |
| OPEN | 0.0394** | 0.0322** | 0.0394** |
| | (1.94) | (1.99) | (1.88) |
| RD | 0.1273*** | 0.1658*** | 0.1273*** |
| | (3.10) | (5.39) | (2.99) |
| constant | -0.0924*** | -0.0378** | -0.0924*** |
| | (-3.47) | (-1.52) | (-3.35) |
| Wald | 1348.78 | 806.43 | - |
| $R^2$ | - | - | 0.9108 |
| N | 120 | 120 | 120 |

above conclusions. The instrumental variable estimation results in this paper are valid, and the robustness of the research hypothesis is further supported.

## 6 Conclusions

As China's economic development has entered a new normal status, the digital economy has become one of the important ways to drive high-quality economic development. Academic circles have not comprehensively discussed the effect and mechanism in the impact of the digital economy on high-quality economic development. For this reason, this paper empirically examines the effect of the digital economy on high-quality economic development, demonstrates its transmission mechanism from the perspective of technological progress, and draws the following main conclusions.

From 2015 to 2019, the development level of China's digital economy had been showing an increasing trend year by year, in which the increase in the sub-index of the digital infrastructure is more obvious. Besides, the growth trend from the digital fusion effect has been basically consistent with the development level of the digital economy, but the development of digital industry has been relatively slow. Therefore, it is necessary to vigorously develop the digital industry. Attention should not only be paid to the development of the consumer Internet but a focus should also be placed on the digital transformation of the industrial Internet and the real economy; the digital industry must be gradually released to lead the digital economy and high-quality economic development. Firstly, the construction of the digital infrastructure should be consolidated, the deep integration of new infrastructure, such as 5G, big data, the Internet of Things, artificial intelligence and industrial networks, should be actively promoted, and traditional enterprises should be supported to carry out sustainable digital infrastructure construction and transformation. Secondly, China should deeply cultivate the development of the industrial Internet and promote the construction of a modern industrial system. It is necessary to establish a perfect cross-boundary capacity and mechanism of the industrial Internet and to

create a good environment for cooperation between physical enterprises and Internet enterprises by breaking down various industrial, regional and operational barriers. A technical system, standards and norms, business models and competition rules must be established and adapted in an integrated development.

The development level of the digital economy in eastern China is much higher than that in other non-eastern regions, and the development of the digital economy in Beijing, Guangdong, Shanghai, Jiangsu and Zhejiang is in the first echelon of the country. In addition, in the eastern region, the marginal contribution of the digital economy to the improvement of total factor productivity is significantly higher than that in the non-eastern region, which has much to do with the first-mover advantage and comparative advantage in the development of digital economy in the eastern region. From the perspective of sub-dimensions, regarding the role of regional heterogeneity in the effects of the digital economy's promotion of high-quality economic development, the influence coefficient of the digital infrastructure sub-index is the largest, followed by the industrial sub-index and finally the integration sub-index. This shows that there is an obvious "digital access gap" between the eastern region and the non-eastern region, while the "digital industry gap" is the smallest among regions. It is necessary to narrow the "digital economy gap" between regions. While the country is expanding the size of the digital economy as a whole, all regions should develop a digital economy industry according to local conditions and should highlight the comparative advantages of the digital economy industry in each region. Firstly, on the basis of ensuring the key links and core enterprises of the digital economy industrial chain in the eastern region, China should promote the gradient transfer of digital economic production capacity in developed areas to the central, western and northeastern regions in an orderly manner. The digital economic development experience of developed regions should be replicated and spread to other regions. Secondly, the government's governance system should be improved, and the modernization level of national digital governance should be enhanced. The empirical results of this paper show that financial R&D subsidies have no significant positive effect on the improvement of technological progress and the development level of the digital economy and even affect the market competition mechanism and lead to inefficiency. Therefore, it is necessary to clarify the position of the government in leading the development of the digital economy, to draw a clear distinction between the market and enterprises, to formulate a negative control list of financial R&D investment, and to accurately subsidize basic and nonmarket R&D. The financial subsidies that affect market competition should be strictly controlled.

The comprehensive index of the digital economy, the sub-index of digital infrastructure, the sub-index of the digital industry and the sub-index of digital fusion all have a significant positive impact on regional total factor productivity, which confirms the importance of developing a digital economy in order to achieve high-quality economic development. The influence coefficients of the three sub-indices ranked from large to small are as follows: the integration sub-index, the infrastructure sub-index and the industry sub-index. In addition, technological progress plays an important role in the influence mechanism of the digital economy on high-quality economic development, and its intermediary effect can explain approximately 1/3 of the total effect. In this regard, it is necessary to highlight the role of technological progress, attach importance to technological innovation at the bottom of the digital layer, and to facilitate sustainable innovation and development. First, Break through the core technological bottleneck with a new strategy of open thinking and comprehensive endogenous capabilities for independent innovation, and achieve a dynamic balance between core independent control and opening up. Second, based on the strategy of expanding domestic demand, China should highlight the social requirements for the development of digital core technology, break the institutional and institutional obstacles that hinder the circulation of talent, capital, and

other innovative elements in the country, and rely on high-tech enterprises. China should further promote the integration of industry, university and research and solve the transformation from "science" to "technology" and then to "application".

## Supporting information

**S1 Data.**
(XLSX)

**S1 Appendix. The process of data standardization and weight determination.**
(DOCX)

## Author Contributions

**Conceptualization:** Siqi Zhao.

**Data curation:** Wei Zhang.

**Formal analysis:** Xiaoyu Wan.

**Methodology:** Yuan Yao.

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
