## [Decision Letter · Decision Letter 0]

11 May 2021

PONE-D-21-12870

Study on the effect of digital economy on high-quality economic development in China

PLOS ONE

Dear Dr. Yanqing Luo,

Thank you for submitting your manuscript to PLOS ONE. After careful consideration, we feel that it has merit but does not fully meet PLOS ONE’s publication criteria as it currently stands. Therefore, we invite you to submit a revised version of the manuscript that addresses the points raised during the review process.

We look forward to receiving your revised manuscript.

Kind regards,

László VASA, PhD

Academic Editor

PLOS ONE

Journal Requirements:

 [NO].

5. Please ensure that you refer to Figure 1 in your text as, if accepted, production will need this reference to link the reader to the figure.

6. We note you have included a table to which you do not refer in the text of your manuscript. Please ensure that you refer to Table 4 and 5 in your text; if accepted, production will need this reference to link the reader to the Table.

Reviewers' comments:

Reviewer's Responses to Questions

**Comments to the Author**

1. Is the manuscript technically sound, and do the data support the conclusions?

Reviewer #1: Yes

Reviewer #2: Yes

Reviewer #3: Partly

2. Has the statistical analysis been performed appropriately and rigorously? 

Reviewer #1: Yes

Reviewer #2: I Don't Know

Reviewer #3: Yes

3. Have the authors made all data underlying the findings in their manuscript fully available?

Reviewer #1: Yes

Reviewer #2: Yes

Reviewer #3: Yes

4. Is the manuscript presented in an intelligible fashion and written in standard English?

Reviewer #1: Yes

Reviewer #2: Yes

Reviewer #3: Yes

5. Review Comments to the Author

Reviewer #1: This research examined the dynamic changes of the digital economy in China. The digitalized economy is a new approach, but this research collapses to measure the current variables used in discussing the digital economy. The precise measurements reflect the accurate discussion relating to the literature and theories. For example, the total factor productivity index used as an independent variable reflects the conventional measurement of GDP, which does not describe the rise of the added value of the digital economy. The common problem is that the sharing of the digital economy is not captured yet by the traditional GDP. The authors try to promote a novelty approach in examining the dynamic changes of the digital economy in China, so they should carefully determine the measurement of the variables. In addition, the authors implemented the Solow Model growth as the basis for the development of hypotheses. In my opinion, if the authors appoint the Solow growth model so they should measure the changes in the level of output of TFP in the case of digital economy activities. They implement the C-D function to measure Inter-provincial Total Factor Productivity, and they support the empirical theories and previous results to show the TFP of the digital economy. Does the conventional TFP reflect the shared digital economy of TFP? How do the authors measure? Please support the empirical findings.

The authors also implement Digital Economic Development Index (DEI) as an independent variable, but they do not explain and support previous findings in the literature review. Do the DEI indices reflect the level of capital based on the Solow Model? How to justify in the literature review?

The authors promote the control variables in the model. However, the measurements do not reflect the digitalization of economic activities. For example, the human capital level is measured with the average number of students in colleges and universities per population. How does the previous measurement reflect the employment creation in the digital economy? The comments are similar in implementing general measurements of FIA, IS, OPEN, and RD in contributing to the growth of the digital economy growth.

It is not described what software it was used with. The other is the disturbance caused by the alpha parameter. This is because the alpha parameter is included in the Cobb-Douglas (CD) function in Section 4.1 (I note incidentally that the formula was not numbered, though it should be), and is included in formulas ([Disp-formula pone.0257365.e002]) and ([Disp-formula pone.0257365.e005]) and later in the article confused as to which they are referring to. It would be advisable to use a different notation. It was not clear to me how the C-D function was applied, how it relates to the model denoted by formula ([Disp-formula pone.0257365.e004]). It would be advisable to use the name Cobb-Douglas instead of C-D first and then the abbreviation.

The English writing needs improvement.

The literature review does not support the determination and measurement of the variables comprehensively.

The manuscript should be even more sufficiently supported by evidence or proper references to work done elsewhere. For this purpose authors could downloads the following published article: https://doi.org/10.3390/su12114674, https://doi.org/10.1080/13675567.2017.1393506.

The conclusion does not extend the literature review in term of previous debates and theories.

The citation in the text is not harmonized with the numbering style in the reference.

Reviewer #2: The paper provides a sound analysis of the development of Chinese digital economy and it gives a clear comprehensive picture of what is taking place in this field. It also vividly characterizes differences in the Chinese economy regarding digitalization. At the same time, it would be important to know more about the international context of China's effort to move forward in this field, whether the process of digitalization in China is affected by the Digital Silk Road (BRI) and other additional policies ('dual circulation' promoted by Xi Jinping) and if yes, then how.. I am aware this is not the main topic of the paper, but that would be an important 'takeaway' here

Reviewer #3: The topic the authors analyzed is absolutely actual and relevant, especially in the context of China, which country is the World's number one forerunner in this field.

The article has a massive theoretical foundation; the authors selected the appropriate methodological toolset and a well-established database for their research. Not only the methodology but also the material of the study is provided well. The authors did not formulate a separate methodology chapter, but it is acceptable from my side (usually, I am pretty strict in this regard, but I like the provided structure).

However, the literature review should be established as now it is not clear where and how the authors made it. The theoretical foundation chapters could be part of a literature review chapter; however, an objective, critical, analytical, and comprehensive literature review should be made too, where the existing essential international literature could be reviewed and analyzed.

6. PLOS authors have the option to publish the peer review history of their article (what does this mean?). If published, this will include your full peer review and any attached files.

Reviewer #1: No

Reviewer #2: **Yes: **Csaba Moldicz

Reviewer #3: No

---

## [Author Response · Author response to Decision Letter 0]

12 Jul 2021

Dear Editors and Reviewers:

Thank you for your letter and for the comments concerning our manuscript entitled “Study on the effect of digital economy on high-quality economic development in China” (No.: PONE-D-21-12870). Those comments are all valuable and very helpful for revising and improving our paper, as well as the important guiding significance to our researches. We have studied comments carefully and have made correction which we hope meet with approval. Revised portion are marked in red in the paper.

The main corrections in the paper and the responds to the comments are as following:

1. Response to comment: (Please ensure that your manuscript meets PLOS ONE’s style requirements, including those for file naming.)

Response: We have formatted the manuscript and file naming in accordance with the PLOS ONE’s style requirements.

2. Response to comment: (Please review your reference list to ensure that it is complete and correct.)

Response: We carefully checked the reference list and ensured that it is complete and correct, and arranged them in the order in which they appeared in the text.

3. Response to comment: (About financial disclosure)

Response: This work was supported by the Project of Humanities and Social Sciences Ministry of Education in China [18YJC790224], the Ministry of Education Layout Foundation of Humanities and Social Sciences [19XJA630004], the Planning Project of Chongqing Social Science [2016BS057]，the talent Introduction Project of Chongqing University of Posts and Telecommunications [A2016-04，K2015-128]. The funders play a guiding and supporting role in our study, and they had no role in study design, data collection and analysis, decision to publish, or preparation of the manuscript.

4. Response to comment: (Ensure that you refer to Figure 1, Table 4 and Table 5 in your text.)

Response: We are very sorry for our negligence that Figure 1, Table 4 and Table 5 are missing in text. In this regard, we refer to Figure 1, Table 1 and Table 2 in the text of manuscript, and added instructions to the manuscript.

Responds to the reviewers’ comments:

Reviewer #1

1. Response to comment: (This research collapses to measure the current variables used in discussing the digital economy.)

Response: Regarding the measurement of the relevant variables of the digital economy, in the second section of the article, we constructed an indicator system for measuring the development of the digital economy. Firstly, we define the definition of the digital economy, and then refer to the existing research to determine the secondary indicators of the system. Secondly, we combine the connotation of the digital economy and related indicator systems in the introduction, and build a measurement system from three secondary indicators of digital infrastructure, digital industry, and digital integration. Finally, on the basis of defining and understanding each second-level index, the third-level index is selected based on the availability of data.

2. Response to comment: (About TFP. The total factor productivity index does not describe the rise of the added value of the digital economy. The authors should measure the changes in the level of output of TFP in the case of digital economy activities. Does the conventional TFP reflect the shared digital economy of TFP? How do the authors measure?)

Response: High-quality economic development is indeed a concept with a wide range of dimensions. It is not only reflected in the efficiency of economic development, but also in many aspects such as innovation-driven, coordinated development, structural optimization, and green development. However, in view of the fact that the research object of this article is the economic development efficiency in the concept of high-quality economic development, that is, focusing on the efficiency part of high-quality economic development, a single indicator (total factor productivity) is used to represent high-quality economic development, which is in line with the practice of many scholars (Siming Liu, 2019; Yueyou Zhang, 2018; Shiyi Chen, 2018). In addition, we have also tried to construct an evaluation index system for high-quality economic development, but in this way, studying the impact of the digital economy measurement system on the high-quality economic development will inevitably lead to structural differences between independent variables and dependent variables, which will affect the research results. Therefore, based on the research object and scope of this article, we use total factor productivity (TFP) to represent high-quality economic development.

3. Response to comment: (It is not described what software it was used with.)

Response: We use Excel office software for data statistics and Stata 15.0 software for regression analysis, regional heterogeneity analysis, robustness test and endogenous test. We added this part in section 4.1 of the manuscript.

4. Response to comment: (Use the name Cobb-Douglas instead of C-D first and then the abbreviation.)

Response: We changed the abbreviation (C-D) to the full name (Cobb-Douglas) of in the manuscript.

5. Response to comment: (The English writing needs improvement.)

Response: In order to improve English writing, we carefully checked the manuscript and made serious changes to the incorrect grammar and inappropriate description.

6. Response to comment: (Regarding the control variables in the model.)

Response: We refer to some previous studies related to the digital economy and make selections based on the purpose of the research. We summarize the potential factors or conditions that can affect the results of the experiment in addition to the experimental factors, and finally obtain these control variables in the model, and these control variables have certain theoretical support. According to the reviewer’s comments, we have modified some of the control variables, using the proportion of technological R&D investment of industrial enterprises in regional GDP to represent the level of regional technological development, and the proportion of government technology R&D subsidies in the government’s general budget expenditure to represent the government’s investment in technology R&D, so that some variables can better meet the digital requirements. At the same time, in the process of revising the paper and searching for relevant data, we found that the data has been updated to 2019. To ensure the timeliness of this research, we have also updated the data during the revision process.

7. Response to comment: (The citation in the text is not harmonized with the numbering style in the reference.)

Response: We carefully checked the reference list and arranged them in the order in which they appeared in the text, and ensured that the citation in the text is harmonized with the numbering style in the reference.

8. Response to comment: (The formula Cobb-Douglas function was not numbered, though it should be.)

Response: We are so sorry for our negligence, and we have added a number to the Cobb-Douglas function.

9. Response to comment: (The authors implement Digital Economic Development

Index as an independent variable, they do not explain and support previous findings in the literature review.)

Response: According to the comment, we have added relevant research to the second part of the article as a theoretical derivation of the main effect of the development of the digital economy in promoting high-quality economic development. In order to measure the development level of the digital economy more comprehensively, we constructed a digital economy development index system in the third part to measure independent variables.

Reviewer #2

1. Response to comment: (It would be important to know more about the international context of China’s effort to move forward in this field, whether the process of digitalization in China is affected by the Digital Silk Road (BRI) and other

additional policies (‘dual circulation’ promoted by Xi Jinping), which would be an important ‘takeaway’ here)

Response: According to the suggestions, we have added relevant content about the international background and the new development pattern of the domestic and international dual cycle in the introduction part of the article.

Reviewer #3

1. Response to comment: (The literature review should be established as now it is

not clear where and how the authors made it.)

Response: Our literature review is included in the introduction to sort out the connotation of the digital economy and current related research on the digital economy. At the same time, according to the reviewer’s suggestion, we added a theoretical analysis part, through literature review, to provide theoretical support for the article’s model framework.

Special thanks to you for your good comments.

In the process of revising the paper and searching for relevant data, we found that the data has been updated to 2019. To ensure the timeliness of this research, we have also updated the data during the revision process and revised Fig 1.

We tried our best to improve the manuscript and made some changes in the manuscript. These changes will not influence the content and framework of the paper.

We appreciate for Editors/Reviewers’ warm work earnestly, and hope that the correction will meet with approval.

Once again, thank you very much for your comments and suggestions.

Best regards,

Wei Zhang, Siqi Zhao, Xiaoyu Wan, Yuan Yao

---

## [Decision Letter · Decision Letter 1]

31 Aug 2021

Study on the effect of digital economy on high-quality economic development in China

PONE-D-21-12870R1

Dear Dr. Zhao,

We’re pleased to inform you that your manuscript has been judged scientifically suitable for publication and will be formally accepted for publication once it meets all outstanding technical requirements.

Kind regards,

László Vasa, PhD

Academic Editor

PLOS ONE

Additional Editor Comments (optional):

Reviewers' comments:

Reviewer's Responses to Questions

**Comments to the Author**

1. If the authors have adequately addressed your comments raised in a previous round of review and you feel that this manuscript is now acceptable for publication, you may indicate that here to bypass the “Comments to the Author” section, enter your conflict of interest statement in the “Confidential to Editor” section, and submit your "Accept" recommendation.

Reviewer #1: All comments have been addressed

2. Is the manuscript technically sound, and do the data support the conclusions?

Reviewer #1: Yes

3. Has the statistical analysis been performed appropriately and rigorously? 

Reviewer #1: Yes

4. Have the authors made all data underlying the findings in their manuscript fully available?

Reviewer #1: Yes

5. Is the manuscript presented in an intelligible fashion and written in standard English?

Reviewer #1: Yes

6. Review Comments to the Author

Reviewer #1: The authors have made necessary improvements to the original submission based on the first review. More sections have been reorganized. Finally, the authors corrected all the spelling errors and rephrased sentences as much as possible in the resubmitted manuscript. After analyzing the review of the manuscript as well as all the changes already introduced during the revision process, I am convinced that the review has improved the quality of the paper since the original submission. In fact, the manuscript has addressed all the reviewer’s comments. On the basis of these observations this manuscript provides sufficient contribution to accept for publication.

7. PLOS authors have the option to publish the peer review history of their article (what does this mean?). If published, this will include your full peer review and any attached files.

Reviewer #1: No

---

## [Editor Report · Acceptance letter]

4 Sep 2021

PONE-D-21-12870R1 

Study on the effect of digital economy on high-quality economic development in China 

Dear Dr. Zhao:

I'm pleased to inform you that your manuscript has been deemed suitable for publication in PLOS ONE. Congratulations! Your manuscript is now with our production department. 

Kind regards, 

on behalf of

Prof. Dr. László Vasa 

Academic Editor

PLOS ONE